# New Insight of Variance reduce in Zero-Order Hard-Thresholding: Mitigating Gradient Error and Expansivity Contradictions

**Xinzhe Yuan**[1], **William de Vazelhes**[2], **Bin Gu**[2,3*], **Huan Xiong**[1,2*]

[1] IASM, Harbin Institute of Technology
[2] Mohamed bin Zayed University of Artificial Intelligence
[3] School of Artificial Intelligence, Jilin University
22s012015@stu.hit.edu.cn
{william.vazelhes, bin.gu}@mbzuai.ac.ae
huan.xiong.math@gmail.com

## Abstract

Hard-thresholding is an important type of algorithm in machine learning that is used to solve $\ell_0$ constrained optimization problems. However, the true gradient of the objective function can be difficult to access in certain scenarios, which normally can be approximated by zeroth-order (ZO) methods. The SZOHT algorithm is the only algorithm tackling $\ell_0$ sparsity constraints with ZO gradients so far. Unfortunately, SZOHT has a notable limitation on the number of random directions due to the inherent conflict between the deviation of ZO gradients and the expansivity of the hard-thresholding operator. This paper approaches this problem by considering the role of variance and provides a new insight into variance reduction: mitigating the unique conflicts between ZO gradients and hard-thresholding. Under this perspective, we propose a generalized variance reduced ZO hard-thresholding algorithm as well as the generalized convergence analysis under standard assumptions. The theoretical results demonstrate the new algorithm eliminates the restrictions on the number of random directions, leading to improved convergence rates and broader applicability compared with SZOHT. Finally, we illustrate the utility of our method on a ridge regression problem as well as black-box adversarial attacks.

## 1 Introduction

$\ell_0$ constrained optimization is a fundamental method in large-scale machine learning, particularly in high-dimensional problems. This approach is widely favored for achieving sparse learning. It offers numerous advantages, notably enhancing efficiency by reducing memory usage, computational demands, and environmental impact. Additionally, this constraint plays a crucial role in combatting overfitting and facilitating precise statistical estimation (Negahban et al., 2012; Raskutti et al., 2011; Bühlmann and Van De Geer, 2011; Yuan and Li, 2021). In this study, we focus on the following problem:

$$\min_{\theta \in \mathbb{R}^d} \mathcal{F}(\theta) = \frac{1}{n} \sum_{i=1}^{n} f_i(\theta), \quad s.t. \ \|\theta\|_0 \leq k, \tag{1}$$

Here, $\mathcal{F}(\theta)$ is the (regularized) empirical risk. $\|\theta\|_0$ represents the number of non-zero directions. $d$ is the dimension of $\theta$. Unfortunately, due to the $\ell_0$ constraint, (1) becomes an NP-hard problem, rendering traditional methods unsuitable for its analysis.

Therefore, we consider using the hard-threshold iterative algorithm (Raskutti et al., 2011; Jain et al., 2014; Nguyen et al., 2017b; Yuan et al., 2017), which is a widely used technique for obtaining approximate solutions to NP-hard's $\ell_0$ constrained optimization problems. Specifically, this technique alternates between the gradient step and the application of the hard threshold operator $\mathcal{H}_k(\theta)$. Operator

---

[*]Corresponding authors.

$\mathcal{H}_k(\theta)$ retains the top $k$ elements of $\theta$ while setting all other directions to zero. The advantage of hard-thresholding over its convex relaxations is that it can achieve similar precision without the need for computationally intensive adjustments, such as tuning $\ell_1$ penalties or constraints. Hard-thresholding was first used for its full gradient form(Jain et al., 2014). Nguyen (Nguyen et al., 2017b) developed a stochastic gradient descent SGD version of hard thresholding known as StoIHT. Nevertheless, StoIHT's convergence condition is overly stringent for practical applications(Li et al., 2016). To address this issue, (Zhou et al., 2018), (Shen and Li, 2017) and (Li et al., 2016) implemented variance reduction techniques to improve the performance of StoIHT in real-world problem-solving.

However, this type of stoIHT is still not suitable for many problems. For example, in certain graphical modeling tasks (Blumensath and Davies, 2009), obtaining the gradient is computationally hard. Even worse, in some settings, the gradient is inaccessible by nature, for instance in bandit problems (Shamir, 2017), black-box adversarial attacks(Tu et al., 2019; Chen et al., 2017; 2019), or reinforcement learning (Salimans et al., 2017; Mania et al., 2018; Choromanski et al., 2020). To address these challenges, zeroth-order (ZO) optimization methods have been developed(Nesterov and Spokoiny, 2017). These methods commonly replace the inaccessible gradient with its finite difference approximation which can be calculated by simply using the function evaluations. Subsequently, ZO methods have been adapted to handle convex constraint sets, rendering them suitable for solving the $\ell_1$ convex relaxation of the problem (1)(Liu et al., 2018; Balasubramanian and Ghadimi, 2018). However, it's essential to highlight that in the context of sparse optimization, $\ell_1$ regularization or constraints can introduce substantial estimation bias and result in inferior statistical properties when compared to $\ell_0$ regularization and constraints(Fan and Li, 2001; Zhang, 2010).

To tackle this issue, a recent development introduced the Stochastic Zeroth-Order Hard-Thresholding algorithm (SZOHT)(de Vazelhes et al., 2022), specifically designed for $\ell_0$ sparsity constraints and gradient-free optimization. Unfortunately, as the only available algorithm in zeroth-order hard-thresholding so far, SZOHT has notable limitations due to the inherent conflict between the deviation of ZO estimators and the expansivity of the hard-thresholding. This limitation makes the algorithm difficult to use in practice, and a natural question is proposed: Could we have a simple ZO hard-thresholding algorithm whose convergence does not rely on the number of $q$ (the number of random directions used to estimate the gradient, further defined in Section 2)?

In this paper, we provide a positive response to this question. Our approach centers on the role of variance in addressing this problem. We firmly believe that variance reduction can offer a dual benefit. It not only holds the potential to accelerate convergence speed but, more importantly, it can effectively mitigate the unique conflicts associated with zero-order hard-thresholding. From this perspective, SZOHT is characterized by its limitation in restricting the sampling of zero-order gradients, essentially representing an incomplete approach to variance reduction. This incompleteness leads to strict conditions for SZOHT. In contrast, we have developed better algorithms by using historical gradients to reduce variance thoroughly. We then provide the convergence and complexity analysis for the generalized variance reduce algorithm under the standard assumptions of sparse learning, which are restricted strong smoothness (RSS), and restricted strong convexity (RSC) (Nguyen et al., 2017b; Shen and Li, 2017) to retain generality. These algorithms eliminate the restrictions on zero-order gradient steps, leading to improved convergence rates and broader applicability. Crucial to our analysis is to provide how variance reduction mitigates contradictions on the parameters $q$ and $k$. Finally, we demonstrate the effectiveness of our method by applying it to both ridge regression problems and black-box adversarial attacks. Our results highlight that our method can achieve competitive performance when compared to state-of-the-art methods for zeroth-order algorithms designed to enforce sparsity.

The majority of our work can be summarized in three parts:

1. New Perspective on Resolving Conflicts Between Zeroth-Order Methods and Hard-Thresholding. Our paper acknowledges the necessity of mitigating this contradiction, emphasizing the demand for a more flexible and resilient approach. By employing the perspective of variance to analyze this issue, our paper presents a more practical and effective solution.

2. Variance Reduction: Another key innovation presented in the paper is the introduction of variance reduction. This concept provides a unique solution to $\ell_0$-constrained zeroth-order optimization. By employing data-driven techniques to reduce variance, the paper not only

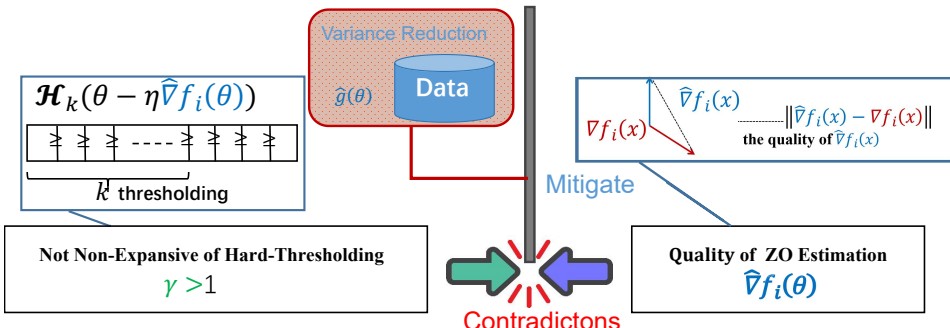

Figure 1: Motivation of our algorithm.

enhances the algorithm's convergence but also expands its utility across a wider range of scenarios.

3. General Analysis: The introduction of a general analysis framework is another contribution to the paper. This framework systematically evaluates the performance and behavior of varying variance reduced algorithms under $\ell_0$-constraint and ZO gradient.

## 2 PRELIMINARIES

Throughout this paper, we use $\|\theta\|$ to denote the Euclidean norm for a vector, $\|\theta\|_\infty$ to denote the maximum absolute component of that vector, and $\|\theta\|_0$ to denote the $\ell_0$ norm (which is not a proper norm). The following two assumptions are widely adopted (Li et al., 2016; Nguyen et al., 2017b) and are needed in this paper.

**Assumption 1** (Restricted strong convexity (RSC) (Li et al., 2016; Nguyen et al., 2017b)). *A differentiable function $\mathcal{F}$ is restricted $\rho_s^-$-strongly convex at sparsity $s$ if there exists a generic constant $\rho_s^- > 0$ such that for any $\theta, \theta' \in \mathbb{R}^d$ with $\|\theta - \theta'\|_0 \leq s$, we have:*

$$\mathcal{F}(\theta) - \mathcal{F}(\theta') - \langle \nabla\mathcal{F}(\theta'), \theta - \theta'\rangle \geq \frac{\rho_s^-}{2}\|\theta - \theta'\|_2^2. \tag{2}$$

**Assumption 2** (Restricted strong smoothness (RSS) (Li et al., 2016; Nguyen et al., 2017b)). *For any $i \in [n]$, a differentiable function $f_i$ is restricted $\rho_s^+$-strongly smooth at sparsity level $s$ if there exists a generic constant $\rho_s^+ > 0$ such that for any $\theta, \theta' \in \mathbb{R}^d$ with $\|\theta - \theta'\|_0 \leq s$, we have*

$$\|\nabla f_i(\theta) - \nabla f_i(\theta')\| \leq \rho_s^+\|\theta - \theta'\|.$$

We assume that the objective function $\mathcal{F}(\theta)$ satisfies the RSC condition and that each component function $f_i(\theta)_{i=1}^n$ satisfies the RSS condition. We also define the restricted condition number as $\kappa_s = \rho_s^+/\rho_s^-$. This assumption ensures that the objective function behaves like a strongly convex and smooth function over a sparse domain, even when it is non-convex.

### 2.1 ZO ESTIMATE

Then, we give our zeroth-order gradient estimator below adopted by (de Vazelhes et al., 2022):

$$\hat{\nabla}f(\theta) = \frac{d}{q\mu}\sum_{i=1}^{q}(f(\theta + \mu\boldsymbol{u}_i) - f(\theta))\boldsymbol{u}_i, \tag{3}$$

where each random direction $\boldsymbol{u}_i$ is a unit vector sampled uniformly from the set $\{\boldsymbol{u} \in \mathbb{R}^d : \|\boldsymbol{u}\|_0 \leq s_2, \|\boldsymbol{u}\| = 1\}$, $q$ is the number of random unit vectors, and $\mu > 0$ is a constant called the *smoothing radius* (typically taken as small as possible, but no too small to avoid numerical errors). To obtain these vectors, we can first sample a random set of coordinates $S$ of size $s_2$ from $[d]$. Following, we sample a random vector $\boldsymbol{u}$ supported on $S$, in other words, uniformly sampled from the set $\{u \in \mathbb{R}^d : \boldsymbol{u}_{[d]-S} = \boldsymbol{0}, \|\boldsymbol{u}\| = 1\}$. Especially, if $s_2 = d$, the general estimator is the usual vanilla estimator with uniform smoothing on the sphere (Gao et al., 2018). Additionally, for convenience, we

define $\mathcal{I}^* = supp(\theta^*)$ as the support of $\theta^*$. Let $\theta^{(r)}$ be a sparse vector with $\|\theta^{(r)}\|_0 \leq k$ and support $\mathcal{I}^{(r)} = supp(\theta^{(r)})$. Define, with $\mathcal{H}_{2k}(\cdot)$ the hard-thresholding operator of sparsity $2k$:

$$\widetilde{\mathcal{I}} = supp(\mathcal{H}_{2k}(\hat{\nabla}\mathcal{F}(\theta^*)) \cup supp(\theta^*).$$

and let $\mathcal{I} = \mathcal{I}^{(r)} + \mathcal{I}^{(r+1)} + \widetilde{\mathcal{I}}$. $\varepsilon_\mu = \rho_s^{+2}sd$, $\varepsilon_{\mathcal{I}} = \frac{2d}{q(s_2+2)}\left(\frac{(s-1)(s_2-1)}{d-1} + 3\right) + 2$, $\varepsilon_{\mathcal{I}^c} = \frac{2d}{q(s_2+2)}\left(\frac{s(s_2-1)}{-1}\right)$, $\varepsilon_{abs} = \frac{2d\rho_s^{+2}ss_2}{q}\left(\frac{(s-1)(s_2-1)}{d-1} + 1\right) + \rho_s^{+2}sd$.

## 2.2 REVISIT OF SZOHT

Based on this assumption and ZO estimation, de Vazelhes proposed the SZOHT algorithm (de Vazelhes et al., 2022). The iteration relationship of this algorithm is:

$$\theta^{(r+1)} = \mathcal{H}_k(\theta^{(r)} - \eta\hat{\nabla}\mathcal{F}(\theta^{(r)}))$$

where $\mathcal{H}_k(\cdot)$ is the hard-thresholding operator and $\hat{\nabla}\mathcal{F}(\theta^{(r)})$ is ZO gradient estimate defined by (3), $\eta$ represents learning rate. SZOHT can address some ZO $\ell_0$-constrained problems under specific conditions. However, it's important to note that the hard-thresholding operator, unlike the projection onto the $\ell_1$ ball, lacks non-expansiveness. Consequently, it has the potential to divert the algorithm's iteration away from the desired solution. To deal with this challenge, SZOHT imposes stringent limitations on both $k$ (hard-thresholding coefficients) and $q$. That is,

$$\left(1 - \frac{\rho_s^{-2}}{(4\varepsilon_{\mathcal{I}} + 1)\rho_s^{+2}}\right)\frac{k^*(4\varepsilon_{\mathcal{I}} + 1)^2\rho_s^{+4}}{\rho_s^{-4}} \leq k \leq \frac{d - k^*}{2}$$

and

- **if** $s_2 > 1$: $q \geq \frac{16d(s_2-1)k^*\kappa^2}{(s_2+2)(d-1)}\left[18\kappa^2 - 1 + 2\sqrt{9\kappa^2(9\kappa^2 - 1) + \frac{1}{2} - \frac{1}{2k^*} + \frac{3}{2}\frac{d-1}{k^*(s_2-1)}}\right]$

- **if** $s_2 = 1$: $q \geq \frac{8\kappa^2 d}{\sqrt{\frac{d}{k^*} + 1}}$

Evidently, these conditions are exceedingly stringent and may not be suitable for numerous real-world problems. Therefore, we urgently need an algorithm with fewer constraints.

## 3 GENERAL ANALYSIS WITH VARIANCE

In this section, we will analyze the random ZO hard-thresholding algorithm from the perspective of variance and provide a positive response to the above questions. These algorithms can be described using the following general iterative expression:

$$\theta^{(r+1)} = \mathcal{H}_k(\theta^{(r)} - \eta\hat{g}^{(r)}(\theta^{(r)})), \tag{4}$$

where $\hat{g}^{(r)}(\theta^{(r)})$ is the generalized gradient estimate (applicable to all ZO hard-thresholding algorithms). Let $\alpha = 1 + \frac{2\sqrt{k^*}}{\sqrt{k - k^*}}$. Then, we have:

**Theorem 1.** *Assume that each $f_i$ is $(\rho_{s'}^+, s')-RSS$ and that $\mathcal{F}$ is $(\rho_s^-, s)-RSC$. For any stochastic ZO hard-thresholding algorithm capable of expressing its iterative relationships as described in (4), we can establish the following:*

$$\mathbb{E}\|\theta^{(r+1)} - \theta^*\|_2^2 \leq (1 + \eta^2\rho_s^{-2})\alpha\mathbb{E}\|\theta^{(r)} - \theta^*\|_2^2 + \eta^2\alpha\mathbb{E}\|\hat{g}_{\mathcal{I}}^{(r)}(\theta^{(r)})\|_2^2 - 2\eta\alpha\left[\mathcal{F}(\theta^{(r)}) - \mathcal{F}(\theta^*)\right]$$

$$+ \alpha\frac{n^2\varepsilon_\mu\mu^2}{\rho_s^{-2}}$$

$$\tag{5}$$

**Remark 1.** *Differing from the approach in (Yuan et al., 2017; Nguyen et al., 2017b; de Vazelhes et al., 2022), where the convergence inequality is segregated into linear convergence terms*

*(represented as $(1 + \eta^2 \rho^- s^2)\alpha \mathbb{E}||\theta^{(r)} - \theta^*||_2^2$ in (5) and error terms (represented as $\alpha \frac{n^2 \varepsilon \mu \mu^2}{\rho_s^{-2}} - 2\eta\alpha \left[ \mathcal{F}(\theta^{(r)}) - \mathcal{F}(\theta^*) \right]$ in (5), we have introduced the gradient squared term $\eta^2 \alpha \mathbb{E}||\hat{g}^{(r)}(\theta^{(r)})||_2^2$ to elucidate the role of variance better. We can transform (5) into the form of (Yuan et al., 2017; Nguyen et al., 2017b; de Vazelhes et al., 2022) by establishing an upper bound for the gradient squared term, which is often feasible for specific algorithms.*

**Conflict analysis through variance.** It is worth noting that among these three components, only the gradient squared term $\eta^2 \alpha \mathbb{E}||\hat{g}^{(r)}(\theta^{(r)})||_2^2$ encompasses both the hard-thresholding parameter (included by $\alpha$) and the ZO gradient parameter (included by $||\hat{g}^{(r)}(\theta^{(r)})||_2^2$). In essence, this means that the conflict between expansivity and zeroth-order error can be fully encapsulated through the gradient squared term. More importantly, when our attention is directed towards the gradient squared term, we discover that in cases where the gradient estimation is unbiased, we obtain $\mathbb{E}||\hat{g}^{(r)}(\theta^{(r)})||_2^2 = \text{Var}||\hat{g}^{(r)}(\theta^{(r)})||_2 + \left\|\nabla \mathcal{F}(\theta^{(r)})\right\|^2$, which means that $\mathbb{E}||\hat{g}^{(r)}(\theta^{(r)})||_2^2$ only related to the variance of gradient estimation. *This indicates that the conflict between the expansionary of hard-thresholding and ZO error is actually between hard-thresholding and the variance of gradient estimation.* In SZOHT, we have $\hat{g}^{(r)}(\theta^{(r)}) = \hat{\nabla}\mathcal{F}(\theta^{(r)})$. Then, the gradient squared term becomes $\eta^2 \alpha \mathbb{E}||\hat{\nabla}\mathcal{F}(\theta^{(r)})||_2^2$. In this scenario, to guarantee algorithm convergence, it becomes essential to ensure that the gradient squared term remains within a reasonable upper bound. Due to the fact that $\alpha$ is already required to satisfy certain conditions (which are generated by linear convergence terms and error terms), therefore, the sampling method for ZO gradients must be restricted, which leads to a reduction in the variance. However, due to the technique of sampling used to reduce the variance, the limitation on the number $q$ of random directions is introduced into SZOHT.

**Improvement plan.** A natural idea is to use a more comprehensive variance reduction approach instead of only using sampling technique to reduce $\mathbb{E}||\hat{g}^{(r)}(\theta^{(r)})||_2^2$, which could effectively alleviate the conflict between ZO estimation and hard-thresholding, ultimately enabling the design of algorithms with fewer constraints, broader applicability, and enhanced convergence speed. Based on this perspective, we have developed a generalized variance reduction ZO hard-thresholding algorithm that leverages historical gradients. We will provide a detailed explanation of this algorithm in the next section.

## 4 $p$M-SZHT ALGORITHM FRAMEWORK

This section mainly presents the $p$M-SZHT algorithm framework along with its convergence analysis. This framework encompasses the majority of unbiased stochastic variance-reduction ZO hard-thresholding methods, providing a generalized result.Subsequently, we introduce the VR-SZHT algorithm, a special case under this framework. Additionally, we extend our discussion by introducing SARAH-ZHT (please note that the gradient estimate in this algorithm is biased) and providing its convergence analysis in the appendix.

### 4.1 $p$M-SZHT

We now present our generalized algorithm to solve the target problem (1), which we name $p$M-SZHT ($p$ Memorization Stochastic Zeroth-Order Hard-Thresholding). Each iteration of our algorithm is composed of two steps: (i) the gradient estimation step, and (ii) the hard thresholding step, where the gradient estimation step includes the variance reduce estimation and zeroth-order estimation. We give the full formal description of our algorithm in Algorithm (1).

In the gradient estimation step, we are utilizing the $p$-Memorization framework, which was originally proposed by Hofmann (Hofmann et al., 2015) to analyze the sequential stochastic gradient algorithm for convex and smooth optimization problems. It's worth noting that our gradient estimation can be seen as its zeroth-order variant (the zeroth-order estimation is shown in Section 2.2). Here, we select in each iteration a random index set $J \subseteq [n]$ of memory locations to update according to:

$$\forall j \in [n]: \ \hat{a}_j^+ := \begin{cases} \hat{\nabla} f_j(\theta), & \text{if } j \in J \\ \hat{a}_j, & \text{otherwise} \end{cases}$$

such that any $j$ has the same probability of $p/n$ being updated[1], where $p$ is the number of directions updated each time (see (Hofmann et al., 2015)). The value of $p$ set $J$, $\forall j, \sum_{J \ni j} \boldsymbol{P}\{J\} - \frac{p}{n}$. Its probability is determined by some specific algorithm. For example, if $\boldsymbol{P}\{J\} = 1/\binom{n}{p}$ if $|J| = p$, and $\boldsymbol{P}\{J\} = 0$ otherwise, we obtain the $p$-SAGA-ZHT algorithm. If $\boldsymbol{P}\{\emptyset\} = 1 - \frac{p}{n}$ and $\boldsymbol{P}\{[1:n]\} = \frac{p}{n}$, we obtain a variant of the VR-SZHT algorithm from Section 4.2. Those are the ZO hard-thresholding versions of the algorithms mentioned in Hofmann et al. (2015); Gu et al. (2020).

In the hard thresholding step, we only keep the $k$ largest (in magnitude) components of the current iterate $\theta^{(r)}$. This ensures that all our iterates (including the last one) are $k$-sparse. This hard-thresholding operator has been studied for instance in (Shen and Li, 2017), and possesses several interesting properties. Firstly, it can be seen as a projection on the $\ell_0$ ball. Second, importantly, it is not non-expansive, contrary to other operators like the soft-thresholding operator (Shen and Li, 2017).

---

**Algorithm 1** Stochastic variance reduced zeroth-order Hard-Thresholding with $p$-Memorization ($p$M-SZHT)

---

**Input:** Learning rate $\eta$, maximum number of iterations $T$, initial point $\theta^{(0)}$, number of random directions $q$, and number of coordinates to keep at each iteration $k$.

**Output:** $\theta^{(r)}$.

1: **for** $r = 1, \ldots, T$ **do**
2:      Update $\hat{a}^{(r-1)}$
3:      Randomly sample $i_r \in \{1, 2, \ldots, n\}$
4:      $\hat{g}^{(r-1)}(\theta^{(r-1)}) = \hat{\nabla} f_{i_r}(\theta^{(r-1)}) - \hat{a}_{i_r}^{(r-1)} + \frac{1}{n} \sum_{j=1}^{n} \hat{a}_j^{(r-1)}$
5:      $\theta^{(r)} = \mathcal{H}_k(\theta^{(r-1)} - \eta g^{(r-1)}(\theta^{(r-1)}))$
6: **end for**

---

**Convergence Analysis**: We provide the convergence analysis of $p$M-SZHT, using the assumptions from Section 2, and demonstrate the correctness of the conclusions made in Section 3 by assessing whether the algorithm converges independently of $q$.

**Theorem 2.** *Suppose $\mathcal{F}(\theta)$ satisfies the RSC condition and that the functions $\{f_i(\theta)\}_{i=1}^n$ satisfy the RSS condition with $s = 2k + k^*$. For Algorithm 1, suppose that we run SZOHT with random supports of size $s_2$, $q$ random directions, a learning rate of $\eta$, and $k$ coordinates kept at each iteration. We have:*

$$[\mathbb{E}\mathcal{F}(\theta^{(r+1)}) - \mathcal{F}(\theta^*)] \leq \gamma[\mathbb{E}\mathcal{F}(\theta^{(r)}) - \mathcal{F}(\theta^*)] + 2L_\mu + L_r \tag{6}$$

*here $L_\mu = \alpha \frac{n^2 \varepsilon_\mu \mu^2}{\rho_s^{-2}} + 6\alpha \varepsilon_{abs} \mu^2 + 6\eta^2 \alpha A_r$, $L_r = \sqrt{s}||\nabla \mathcal{F}(\theta^*)||_\infty \mathbb{E}||\theta^{(r)} - \theta^*||_2 + \eta^2(3\alpha((4\varepsilon_\mathcal{I} s + 2) + \varepsilon_{\mathcal{I}^c}(d - k))\mathbb{E}||\nabla f_{i_t}(\theta^*)||_\infty^2)$, $\gamma = \left(\frac{2\beta}{\rho_s} + 48\eta^2 \alpha \rho_s^+ \varepsilon_\mathcal{I} - 2\eta\alpha + 1 - \frac{p}{n}\right)$.*

**Remark 2.** *(System error). This format of result is similar to the ones in (Yuan et al., 2017; Nguyen et al., 2017b; de Vazelhes et al., 2022), the right of (25) contains a linear convergence term $\gamma[\mathbb{E}\mathcal{F}(\theta^{(r)}) - \mathcal{F}(\theta^*)]$, and system error $2L_\mu + L_r$. We note that if $\mathcal{F}$ has a $k^*$-sparse unconstrained minimizer, which could happen in sparse reconstruction, or with overparameterized deep networks, then we would have $||\nabla \mathcal{F}(\theta^*)||_\infty = 0$ and $||\nabla f_{i_r}(\theta^*)||_\infty^2 = 0$, and hence that part of the system error $L_r$ would vanish. In addition, we also have another system error $L_r$, which depends on the smoothing radius $\mu$, due to the error from the ZO estimate and the iterative method of $\hat{a}$.*

From this theorem, we know that if the algorithm converges, $\eta$ needs to lie in some specific interval.

**Corollary 1.** *If*

$$\eta' - \frac{\sqrt{\Delta}}{2(48\varepsilon_\mathcal{I} \alpha \rho_s^+ + \rho_s^-)} \leq \eta \leq \max\left\{\eta' + \frac{\sqrt{\Delta}}{2(48\varepsilon_\mathcal{I} \alpha \rho_s^+ + \rho_s^-)}, \frac{1}{48\varepsilon_\mathcal{I} \alpha \rho_s^+}\right\} \tag{7}$$

*algorithm 1 converges. Here $\eta' = \frac{\alpha}{48\varepsilon_\mathcal{I} \alpha \rho_s^+ + \rho_s^-}$, $\Delta = 4\alpha^2 - 4(48\varepsilon_\mathcal{I} \alpha \rho_s^+ + \rho_s^-)(1 - \frac{p}{n} + \frac{2}{\rho_s^-})$.*

---

[1]Originally, $p$-Memorization is called $q$-Memorization. We change it to $p$ to avoid conflicting with random directions in zeroth order

**Remark 3.** *(Independence of q) When $k > k^*$, for any $q > 0$ the necessary condition $\Delta > 0$ for (7) holds. We emphasize here that variance reduction can only make q unable to determine whether to converge, but q can still affect the convergence speed. In other words, variance reduction can mitigate the conflict, but cannot resolve it.*

### 4.2 VR-SZHT

To offer a specific analysis, we introduce the VR-SZHT algorithm, which is the adaptation of the original SVRG method Johnson and Zhang (2013) to our ZO hard-thresholding setting. In addition to the previously mentioned convergence analysis, we will also provide a complexity analysis to demonstrate the advantages of this algorithm, which extend beyond existing algorithm.

---

**Algorithm 2** Stochastic variance reduced zeroth-order Hard-Thresholding (VR-SZHT)

---

**Input:** Learning rate $\eta$, maximum number of iterations $T$, initial point $\theta^0$, SVRG update frequency $m$, number of random directions $q$, and number of coordinates to keep at each iteration $k$.

**Output:** $\theta^T$.

 1: **for** $r = 1, \ldots, T$ **do**
 2:     $\theta^{(0)} = \theta^{r-1}$;
 3:     $\hat{\mu} = \frac{1}{n} \sum_{i=1}^{n} \hat{\nabla} f_i(\theta^{(0)})$;
 4:     **for** $t = 0, 1, \ldots, m-1$ **do**
 5:         Randomly sample $i_t \in \{1, 2, \ldots, n\}$;
 6:         Compute ZO estimate $\hat{\nabla} f_{i_t}(\theta^{(r)}), \hat{\nabla} f_{i_t}(\theta^{(0)})$;
 7:         $\bar{\theta}^{(r+1)} = \theta^{(r)} - \eta(\hat{\nabla} f_{i_t}(\theta^{(r)}) - \hat{\nabla} f_{i_t}(\theta^{(0)}) + \hat{\mu})$;
 8:         $\theta^{(r+1)} = \mathcal{H}_k(\bar{\theta}^{(r+1)})$;
 9:     **end for**
10:     $\theta^r = \theta^{(r+1)}$, random $t' \in [m-1]$;
11: **end for**

---

**Theorem 3.** *Suppose $\mathcal{F}(\theta)$ satisfies the RSC condition and that the functions $\{f_i(\theta)\}_{i=1}^{n}$ satisfy the RSS condition with $s = 2k + k^*$. When $\eta = \frac{\alpha \rho_S^-}{2(48\varepsilon_\mathcal{I} \alpha \rho_s^- \rho_s^+ + \rho_s^{-2})}$, we have:*

$$\delta \left[ \mathcal{F}(\widetilde{\theta}^{(r)}) - \mathcal{F}(\theta^*) \right] \leq \gamma' \mathbb{E}[\mathcal{F}(\widetilde{\theta}^{(r-1)}) - \mathcal{F}(\theta^*)] + L'_\mu + L. \tag{8}$$

*Here $\beta = (1 + \eta^2 \rho_s^{-2})\alpha$, $\delta = \frac{\beta^m - 1}{\beta - 1}(2\eta - 48\varepsilon_\mathcal{I}\eta^2\rho_s^+)\alpha$, $\gamma' = (\frac{2\beta^m}{\rho_s^-} + \frac{48\eta^2\rho_s^+\varepsilon_\mathcal{I}\alpha(\beta^m-1)}{\beta-1})$, $L'_\mu = \frac{2\beta^m}{\rho_s^-}\sqrt{s}\|\nabla\mathcal{F}(\theta^*)\|_\infty \mathbb{E}\|\widetilde{\theta}^{(r-1)} - \theta^*\|_2 + 6\eta^2\frac{\beta^m-1}{\beta-1}\alpha((4\varepsilon_\mathcal{I}s+2) + \varepsilon_{\mathcal{I}^c}(d-k))\mathbb{E}\|\nabla f_{i_t}(\theta^*)\|_\infty^2 + 3\|\nabla_\mathcal{I}\mathcal{F}(\theta^*)\|_2^2)$, and $L' = \frac{\beta^m-1}{\beta-1}\alpha(72\eta^2\varepsilon_{abs}\mu^2 + \frac{n^2\varepsilon_\mu\mu^2}{\rho_s^{-2}})$.*

This theorem is similar to Theorem 2. And it is worth noting that $q$ is also independent in VR-SZHT, and can be found in the appendix due to space limitations.

**Corollary 2.** *The ZO query complexity of the algorithm is $\mathcal{O}\left([n + \frac{\kappa^3}{\kappa^2+1}]\log\left(\frac{1}{\varepsilon}\right)\right)$. And the hard-thresholding query complexity is $\mathcal{O}\left(\log(\frac{1}{\varepsilon})\right)$.*

When comparing VR-SZHT with SZOHT, where the ZO query complexity of SZOHT is $\mathcal{O}\left((k + \frac{d}{s_2})\kappa^2 \log(\frac{1}{\varepsilon})\right)$ and the hard-thresholding query complexity is $\mathcal{O}\left(\kappa^2 \log(\frac{1}{\varepsilon})\right)$, it becomes evident that the hard-thresholding query complexity of VR-SZHT is significantly reduced. Furthermore, as $k$ becomes large, the ZO complexity is also reduced.

## 5 EXPERIMENTS

We now compare the performance of VR-SZHT, SAGA-SZHT, and SARAH-SZHT (an adaptation of the SARAH variance reduction method (Nguyen et al., 2017a) to our ZO hard-thresholding setting, for which we provide the convergence analysis in Appendix 6) with that of the following algorithms,

in terms of IZO (iterative zeroth-order oracle, i.e. number of calls to $f_i$) and NHT (number of hard-thresholding operations):

- SZOHT (de Vazelhes et al., 2022): a vanilla stochastic ZO hard-thresholding algorithm.
- FGZOHT: the full gradient version of SZOHT.

**Ridge Regression**    We first consider the following ridge regression problem, where malfunctions $f_i$ are defined as follows: $f_i(\theta) = (x_i^\top \theta - y_i)^2 + \frac{\lambda}{2}\|\theta\|_2^2$, where $\lambda$ is some regularization parameter. We generate each $x_i$ randomly from a unit norm ball in $\mathbb{R}^d$, and a true random model $\theta^*$ from a normal distribution $\mathcal{N}(0, I_{d\times d})$. Each $y_i$ is defined as $y_i = x_i^\top \theta^*$. We set the constants of the problem as such: $n = 10, d = 5, \lambda = 0.5$. Before training, we preprocess each column by subtracting its mean and dividing it by its empirical standard deviation. We run each algorithm with $k = 3, q = 200, \mu = 10^{-4}, s_2 = d = 5$, and for the variance reduced algorithms, we choose $m = 10$. For all algorithms, the learning rate $\eta$ is found through grid-search in $\{0.005, 0.01, 0.05, 0.1, 0.5\}$: we choose the learning rates giving the lowest function value (averaged over several runs) at the end of training. We stop each algorithm once its number of IZO reaches 80,000. All curves are averaged over 3 runs, and we plot their mean and standard deviation in Figure 2. As we can observe, SZOHT converges to higher function values than other algorithms: this illustrates the advantage of the variance reduction techniques, which can allow to attain smaller function values than plain SZOHT, but at a cheaper cost than FGZOHT.

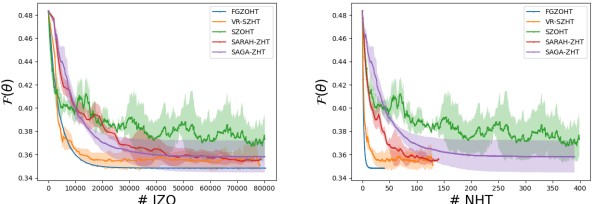

Figure 2: #IZO and #NHT on the ridge regression task.

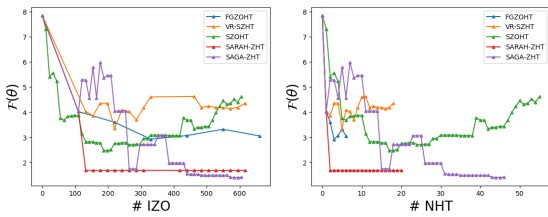

Figure 3: #IZO and #NHT on the few pixels adv. attacks (CIFAR-10), for the original class 'airplane'.

**Few Pixels Universal Adversarial Attacks**    Finally, we consider a few-pixel universal adversarial attacks problem. Let some classifier be trained on a dataset of images. We assume that it can only be accessed as a black box, i.e. it only returns the log probabilities of each estimated class, given an input image. This is a typical real-life scenario, where for instance the model can only be accessed through a provider's API. We seek to find a single perturbation $\theta \in \mathbb{R}^d$, to apply to several images at once, (we denote those images by $x_i$, $i = \{1, \ldots, n\}$, and their true label as $y_i$) to make the predicted class for those images different than their true class. Further discussion on universal perturbations can be found in (Dezfooli et al., 2017). In addition, we seek an adversarial perturbation that is sparse, to preserve as much as possible the original image. As is usual in black-box adversarial attacks, we maximize the following Carlini-Wagner loss (Carlini and Wagner, 2017; Chen et al., 2017), which encourages the prediction from the model to be different from the true class:

$$f_i(\theta) = \max\{F_{y_i}(\text{clip}(x_i + \theta)) - \max_{j\neq y_i} F_j(\text{clip}(x_i + \theta)), 0\},$$

where $x_i$ is the original $i$-th image (rescaled to have values in $[-0.5, 0.5]$), of true class $y_i$, clip denotes the clipping operation into $[-0.5, 0.5]$, $\theta$ is the universal perturbation that we seek to optimize, and

Table 1: Comparison of universal adversarial attacks on $n = 10$ images from the CIFAR-10 test-set, from the 'airplane' class. For each algorithm, the leftmost image is the sparse adversarial perturbation applied to each image in the row. ('auto' stands for 'automobile', and 'plane' for 'airplane')

| Image ID | 3 | 27 | 44 | 90 | 97 | 98 | 111 | 116 | 125 | 153 |
|---|---|---|---|---|---|---|---|---|---|---|
| Original | | | | | | | | | | |
| FGZOHT | plane | plane | plane | **ship** | **deer** | plane | plane | plane | **ship** | **truck** |
| SZOHT | plane | plane | plane | plane | **deer** | plane | **bird** | **bird** | **ship** | **truck** |
| VR-SZHT | plane | plane | **auto** | plane | **ship** | plane | plane | plane | **ship** | **truck** |
| SAGA-SZHT | plane | **frog** | plane | plane | **deer** | plane | plane | plane | **ship** | **ship** |
| SARAH-SZHT | plane | plane | **auto** | **ship** | **ship** | plane | **bird** | **plane** | **frog** | **truck** |

each function $F_k$ outputs the log-probability of image $x_i$ being of class $k$ as predicted by the model, for $k \in \{1, .., K\}$, with $K$ the number of classes (similarly to (Chen et al., 2017; Liu et al., 2018; Huang et al., 2019)). Similarly to Liu et al. (2018) (Appendix A.11), we evaluate the algorithm on a dataset of $n = 10$ images from the test-set of the CIFAR-10 dataset(Krizhevsky et al., 2009), of dimensionality $32 \times 32 \times 3 = 3,072$, from the same class 'airplane', which we display in Table 1. We take as model $F$ a fixed neural network, already trained on the train-set of CIFAR-10, obtained from the supplementary material of (de Vazelhes et al., 2022). We set $k = 60$, $\mu = 0.001$, $q = 10$, $s_2 = d = 3,072$, and the number of inner iterations of the variance reduced algorithms to $m = 10$. We check at each iteration the number of IZO, and we stop training if it exceeds 600. Finally, for each algorithm, we grid-search the learning rate $\eta$ in $\{0.001, 0.005, 0.01, 0.05\}$. The best learning rates (giving the curve which obtained the smallest minimum function value), are respectively: FGZOHT: 0.05, SZOHT: 0.005, VR-SZHT: 0.01, SAGA-SZHT: 0.05, SARAH-SZHT: 0.05. Our experiments are conducted on a workstation of 128 CPU cores. The training curves are presented in Figure 3: SAGA-SZHT obtains the lowest function value at the end of the training, followed by SARAH-SZHT. In terms of attack success rate, SARAH-SZHT presents the highest success rate, as it has successfully attacked 7/10 images. We provide further results, on 3 more classes ('ship', 'bird', and 'dog') in the appendix, which demonstrate even further the advantage of variance reduction methods in our setting.

## 6 CONCLUSION

In this paper, we introduce a novel approach to address sparse zero-order optimization problems and leverage it to enhance existing algorithms. We perform a comprehensive convergence analysis of the generalized variance reduction algorithm, showcasing how variance reduction can effectively mitigate the limitations inherent in existing algorithms. To substantiate our claims, we validate our algorithm through experiments involving ridge regression and adversarial attacks.

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
