# OpenReview forum: "New Insight of Variance reduce in Zero-Order Hard-Thresholding: Mitigating Gradient Error and Expansivity Contradictions"
_ICLR.cc/2024/Conference — ICLR 2024 poster_

### Official Review · Reviewer_7pPP · 2023-10-28

**Soundness:** 3 good
**Presentation:** 3 good
**Contribution:** 3 good
**Rating:** 6
**Confidence:** 2

**Summary:**

This paper studies efficient zeroth order (ZO) algorithms for solving $\ell_0$-constrained optimization problems. While existing approach alternate between gradient descent and hard thresholding, it only works well in restricted settings. This paper makes the novel observation that large variance  of gradient estimation would result in conflict between ZO gradient and expansity of hard-thresholding operators. To resolve this issue, the authors prodive a variance reduced ZO hard-thresholding algorithm, and theoretically demonstrate that the variance-reduced algorithm is guaranteed to converge under standard assumptions and eliminates the requirement of the random direction sampled. Finally, the authors demonstrate the superiority of their algorithm by solving portfolio and adversatial attacks.

**Strengths:**

1) Both the algorithm and the analysis presented in this paper seem novel. Notably, the role of variance discovered by this paper is found by establish a new descent inequality (Theorem 1), and based on this discovery a variance reduced algorithm is introduced.

2) The paper is in general well-written and not hard to read. The authors introduce necessary background of ZO in the introduction section and also discuss previous works, making it easier to understand the contribution of this paper.

3) Besides sound theoretical results, extensive experiments are also conducted to verifty the efficiency of the proposed algorithm.

**Weaknesses:**

1) Some quantities seem to be used before being defined: $k*$ and $\kappa$ in Sec. 2.2, $\hat{g}_I^{(r)}$ in Theorem 1.

2) In the experiments, I think the authors can add more comparison on computational cost and also comparisons with first-order methods or other methods as well, to highlight the benefits of using zeroth order optimization. I'm not an expert in zeroth order optimization, so I think adding such comparisons could make your results more convincing.

**Questions:**

The paper [1] considers a stochastic optimization setting while the current paper studies finite-sum optimization. Can there be a way to reduce the variance for [1]'s setting?

[1] de Vazelhes, W., Zhang, H., Wu, H., Yuan, X., & Gu, B. (2022). Zeroth-Order Hard-Thresholding: Gradient Error vs. Expansivity. Advances in Neural Information Processing Systems, 35, 22589-22601.

---

> ### Author Response · Authors · 2023-11-21
> **Response to Reviewer 7pPP**
>
> Dear reviewer 7pPP, we extend our sincere gratitude for dedicating your valuable time to reviewing our paper and for expressing your appreciation and support for our work. In the following, we will provide a comprehensive response to your review comments.
>
> > **Your comment:** The paper [1] considers a stochastic optimization setting while the current paper studies finite-sum optimization. Can there be a way to reduce the variance for [1]'s setting?
>
>
> **Our response:** Indeed, while [1] considered a general stochastic setting, in our framework, we consider the finite-sum setting, in order to use recent advances in variance-reduction for the finite-sum setting.
> However, we believe it would be hard to reduce the variance *in general*, i.e. for the general stochasticity introduced in [1], since we are unaware of the existence of such technique, even in the much simpler case of vanilla stochastic (non-finite-sum) first order optimization. This is an interesting question however, and we leave this for future work.
>
>
>
>
> > **Your comment:** In the experiments, I think the authors can add more comparison on computational cost and also comparisons with first-order methods or other methods as well, to highlight the benefits of using zeroth order optimization. I'm not an expert in zeroth order optimization, so I think adding such comparisons could make your results more convincing.
>
> **Our response:** Regarding the computational cost, we have added a few precisions regarding the meaning of IZO and NHT in our paper: IZO stands for Iterative Zeroth-Order Oracle, that is, the number of calls to a function $f\_i$, and NHT stands for the number of Hard-Thresholding operations: as such, one can directly use those two metrics to know the computational complexity of the algorithm: for a function $f\_i$ which is costly to evaluate, one would rather seek to minimize the IZO, but for simple functions $f\_i$, the hard-thresholding operation may become the bottleneck and one will then care more about NHT.
> Regarding the benefit of zeroth-order methods compared to first order methods, actually, if one has access to the gradient, then first order methods will always be superior to ZO methods (since ZO gradient estimation introduces a lot of errors, in particular for high-dimensional problems): what we consider in our paper is a case where, due to constraints on the experimental setting (for instance, privacy constraints), one cannot access the gradient of the function, i.e. the function is a black box, which is why we do not compare to first order methods. In other words, we do not study zeroth-order method as an improvement over first-order methods, but rather as a fixed constraint of the problem. In the literature however, some methods inspired by zeroth-order methods, such as [10], can have an advantage in terms of computational memory over first-order methods, but such aspects are beyond the scope of our paper.
>
> [10] David Silver, Learning by directional gradient descent, ICLR, 2021

---

### Official Review · Reviewer_CZzv · 2023-10-29

**Soundness:** 4 excellent
**Presentation:** 2 fair
**Contribution:** 3 good
**Rating:** 6
**Confidence:** 4

**Summary:**

The paper introduces a new algorithm for solving $\ell_0$ constrained optimization problems using zeroth-order (ZO) gradients, specifically targeting the limitations of the SZOHT algorithm, which is currently the only algorithm addressing this problem. The main limitation of SZOHT is the conflict between ZO gradients and the expansivity of the hard-thresholding operator, which restricts the number of random directions. The proposed algorithm takes a novel perspective by considering the role of variance and introduces a generalized variance-reduced ZO hard-thresholding algorithm. The paper provides theoretical analysis and demonstrates that the new algorithm overcomes the limitations of SZOHT, resulting in improved convergence rates and broader applicability. The utility of the proposed method is illustrated through experiments on a ridge regression problem and black-box adversarial attacks.

**Strengths:**

1. The paper presents a well-motivated and principled approach to resolving conflicts between zeroth-order methods and hard thresholding. The authors offer a fresh perspective on the problem, providing innovative insights and potential solutions.
2. The paper introduces a comprehensive analysis framework that evaluates the performance and behavior of variance-reduced algorithms under the $\ell_0$-constraint and zeroth-order gradient settings.
3. The paper includes rigorous theoretical analysis, establishing a solid foundation for the proposed algorithm. The theoretical justifications and proofs contribute to the robustness of the research and build confidence in its practical applicability. Additionally, the paper demonstrates the versatility of the proposed method through applications in ridge regression and black-box adversarial attacks, highlighting its real-world relevance and effectiveness.

**Weaknesses:**

1. The paper lacks a proper related work section, which makes it challenging for readers to quickly grasp the background and understand the previous works. It is crucial to include a comprehensive discussion on related works, especially regarding the variance-reduced ZO hard-thresholding algorithm and the variance reduction aspect.
2. The paper suffers from a lack of necessary references, such as papers on SAGA, SARAH, and SVRG methods. When these methods are initially mentioned, it is essential to provide corresponding references. Additionally, there are errors in the appendix due to bibtex errors, which should be carefully reviewed and corrected.
3. The presentation of baselines and experimental settings in the main text is not well-organized. It is recommended to reorganize this information to improve clarity, especially for readers who are unfamiliar with the baselines and adversarial attacks. Providing a cross-reference to the appendix can also help readers gain a better understanding.
4. The introduction of SAGA-SZHT is missing from the paper, and it cannot be found. It is necessary to either locate the missing information or add it during the rebuttal phase.
5. The authors propose three variants of VR-SZHT by utilizing SVRG, SARAH, and SAGA. It would be beneficial to summarize the advantages of each method in terms of memory storage and convergence rate, similar to what is found in the variance-reduction literature. Providing tables or summaries can help readers compare and understand the individual strengths of these methods.
6. It is well-known that variance-reduction methods can improve the convergence rate of SGD from sublinear to linear under strongly convex and smoothness conditions. It would be interesting to clarify whether VR-SZHT exhibits a similar improvement compared to SZOHT. If there are notable differences, the authors should provide explanations or insights into the reasons behind these variations.
7. I am curious to know if there are any additional technical challenges when integrating VR methods into SZOHT and proving the convergence rate, compared to applying VR methods to traditional finite-sum tasks. The response to this question will not impact my final evaluation of the paper's novelty. However, it will help me gain a better understanding of the paper's correctness and soundness.

**Questions:**

Please refer to the weakness section for detailed feedback. Overall, I appreciate the paper for its natural motivation and principled solutions. However, there are several issues with the presentation that need to be addressed. During the rebuttal phase, I encourage the authors to resolve the mentioned concerns or correct any misunderstandings I may have. I suggest that the authors respond to my questions in multiple phases if they need much time to solve all the issues, allowing for further clarification if needed. For instance, it would be helpful to begin with the related works section and provide the algorithm for SAGA-SZHT as a priority. One suggestion is for the authors to incorporate the appendix into the current PDF version of the paper. This would enhance convenience for reviewers, allowing them to access all relevant information in a single document.

Based on the current version of the paper, I have given it a borderline score. However, I will reconsider my evaluation based on the authors' responses during the rebuttal phase. I would like to defend my positive attitude towards the paper if my concerns and issues are effectively addressed.

---

> ### Author Response · Authors · 2023-11-21
> **Response to Reviewer CZzv [1/2]**
>
> > **Your comment:** The paper lacks a proper related work section, which makes it challenging for readers to quickly grasp the background and understand the previous works. It is crucial to include a comprehensive discussion on related works, especially regarding the variance-reduced ZO hard-thresholding algorithm and the variance reduction aspect. The paper suffers from a lack of necessary references, such as papers on SAGA, SARAH, and SVRG methods. When these methods are initially mentioned, it is essential to provide corresponding references. Additionally, there are errors in the appendix due to bibtex errors, which should be carefully reviewed and corrected.
>
>
>
> Thank you for your comment, it is true that some proper introduction to some of the exposed methods in the paper were missing. We have revised our paper to include references on the methods that we discuss. Additionally, we added an extra section in Appendix which sums up, in a table, all variance reduced algorithms, their original first-order counterpart and computational complexity, and links to the corresponding section in the paper where we analyze them. Finally, thank you for noticing the bibtex errors, we have now corrected them in the new revision.
>
>
>
> > **Your comment:** The presentation of baselines and experimental settings in the main text is not well-organized. It is recommended to reorganize this information to improve clarity, especially for readers who are unfamiliar with the baselines and adversarial attacks. Providing a cross-reference to the appendix can also help readers gain a better understanding.
>
> **Our response:** Thank you for your comment, regarding the baselines, we have revised the paper accordingly to add to the corresponding references from the literature, and wrote an additional section in Appendix to sum-up all our variance reduction algorithms. In the main body of the paper, we have added some cross-references to the corresponding sections from the Appendix (for instance, we linked to the corresponding Appendix section for the proof of SARAH-SZHT). Regarding more precisions about the Adversarial Attacks setting, we have added a reference to the first paper which introduced such a setting: [8] (Appendix 11). We hope this can clarify such experimental setting and the motivations behind it.
>
>
> > **Your comment:** The introduction of SAGA-SZHT is missing from the paper, and it cannot be found. It is necessary to either locate the missing information or add it during the rebuttal phase.
>
>
> **Our response:** As we briefly mention at the end of page 5, SAGA-SZHT is actually a specific instantiation of pM-SZHT, where we only need to let $J \in \\{\\{1\\}, ..., \\{n\\}\\}$ and $p=1$. Thank you for your comment, we agree that this point was not highlighted enough: in the new revision, we also included a bibliographic reference to SAGA at the beginning of page 6 for clarity, and additional references about SAGA in Appendix 1. Additionally, we did not separately provide a proof of convergence for SAGA-SZHT since it falls within the pM-SZHT framework, but we used SAGA as our main example to analyze algorithms within the pM framework.
>
> **[...Continued below...]**

---

> > ### Author Response · Authors · 2023-11-21
> > **Response to Reviewer CZzv [2/2]**
> >
> > **[...Continuing from above...]**
> >
> >
> >
> > > **Your comment:** The authors propose three variants of VR-SZHT by utilizing SVRG, SARAH, and SAGA. It would be beneficial to summarize the advantages of each method in terms of memory storage and convergence rate, similar to what is found in the variance-reduction literature. Providing tables or summaries can help readers compare and understand the individual strengths of these methods. It is well-known that variance-reduction methods can improve the convergence rate of SGD from sublinear to linear under strongly convex and smoothness conditions. It would be interesting to clarify whether VR-SZHT exhibits a similar improvement compared to SZOHT. If there are notable differences, the authors should provide explanations or insights into the reasons behind these variations. I am curious to know if there are any additional technical challenges when integrating VR methods into SZOHT and proving the convergence rate, compared to applying VR methods to traditional finite-sum tasks. The response to this question will not impact my final evaluation of the paper's novelty. However, it will help me gain a better understanding of the paper's correctness and soundness.
> >
> >
> > **Our response:** Thank you for your question. We added an extra section in Appendix 1 which sums up, in a table, all variance reduced algorithms, their original first-order counterpart and computational complexity, and links to the corresponding section in the paper where we analyze them. Regarding the convergence rate, the advantage of variance reduction algorithm over SGD, in the hard-thresholding setting, departs from standard results from convex optimization. Indeed, in the IHT (hard-thresholding) litterature, algorithms optimize the function $\mathcal{F}$ up to some system error (i.e. not to $\varepsilon$ optimality). As such, even the standard SGD-IHT analysis (from Nguyen et. al. 2017) is considered having a linear rate (since SGD, with a constant learning rate, can optimize up to some neighborhood of the optimum (i.e. "up to some system error") with a linear rate). SVRG-HT (Li et. al. 2016), on the other hand, also has a linear rate, but its advantage over SGD-IHT is that it requires less stringent assumptions for convergence: this can be seen for instance in Table 1 from [9], where some assumptions on the condition number of $\mathcal{F}$ can be removed. Therefore, typically, in the IHT setting, improvements of variance-reduction algorithms are not on the convergence rate itself but rather on extending the applicability of the algorithm, as they provide a better way to deal with the variance of the gradient estimation. In our case, variance reduction algorithms improve the applicability of ZO hard-thresholding by removing the stringent conditions on the number of random directions $q$, as we discuss in the paper. Regarding the technical challenge, when integrating VR methods into SZOHT and proving the convergence rate, understanding parameter relationships is vital in convergence analysis, especially in explaining how variance reduction techniques alleviate conflicts. Here, we illustrate this point by comparing SZOHT and VR-SZHT.  In SZOHT, attention is primarily on the constraint link between the zeroth-order estimated direction number $q$ and the Hard Thresholding parameter $k$. However, VR-SZHT introduces variance reduction, adding an inner loop parameter $m$, leading to interdependence among these three parameters. Analyzing this relationship poses a notable challenge. To assess if variance reduction algorithms can mitigate conflicts, we explore whether there exists $m$ to make the algorithm converge for any $q$. Analyzing the coefficients in the convergence term of equation (8) yields inequalities involving $k$,$q$,$m$. Ultimately, confirming a positive outcome is achieved through the existence of $m$ (see Appendix 4.3 for additional insights).
> >
> > [8] Sijia Liu, Zeroth-order stochastic variance reduction for nonconvex optimization. NIPS, 2018
> >
> > [9] Zhou, Pan, Efficient stochastic gradient hard thresholding. NIPS, 2018

---

> > > ### Comment · Reviewer_CZzv · 2023-11-22
> > >
> > > Thank you for your response. I highly recommend that the authors consider incorporating the appendix into the current PDF version of the paper, as permitted by ICLR'24. This would greatly improve convenience for readers, enabling them to access all pertinent information within a single document. After carefully reviewing your feedback and considering the opinions of other reviewers, I maintain a positive attitude towards this paper and have decided to maintain my current rating.

---

> > > > ### Author Response · Authors · 2023-11-22
> > > > **Response to Reviewer CZzv**
> > > >
> > > > Dear reviewer CZzv, thank you for your careful review, and your positive attitude towards our paper. Regarding uploading a single pdf, thank you for your suggestion, we have now uploaded in the pdf field our full paper including appendix as suggested.

---

### Official Review · Reviewer_wexC · 2023-10-30

**Soundness:** 2 fair
**Presentation:** 1 poor
**Contribution:** 2 fair
**Rating:** 3
**Confidence:** 3

**Summary:**

This paper first proposes a general convergence analysis on stochastic zeroth-order hard-thresholding algorithms with variance. Based on the theoretical analysis, this paper further proposes a generalized variance reduced zeroth-order hard-thresholding algorithm. Theoretical results demonstrate the new algorithm has improved convergence rates and broader applicability compared with existing methods by eliminating the restrictions on the number of random directions. Experiments on both ridge regression and black-box adversarial attack demonstrate the effectiveness of proposed method.

**Strengths:**

The idea of combining variance reduction with zeroth-order hard-thresholding methods should be novel

**Weaknesses:**

- The proposed method is not clearly introduced and has many confusing points.
- Experiments is not very supportive for the proposed method, details can be found in Questions part

**Questions:**

- My first concern is on the usefulness of zeroth-order oracles. From experiments in this paper, it seems that we have easy access to gradient information in all these applications. Therefore, why do we need to emphasize zeroth-order algorithms? The authors may consider some other applications where gradient information is hard to obtain.
- Another concern is on the setting of L0 regularization. While there exist many other approaches on obtaining sparse solutions (e.g., LASSO), the authors may need to justify the necessity of using L0 regularization here.
- I am a bit confused on the key contribution of this work. The authors introduced two algorithms, pM-SZHT and VR-SZHT. What is the key difference of these two algorithms? Moreover, pM-SZHT is not compared in experiments, but SARAH/SAGA-SZHT is compared. This is also very confusing and may need some explanations.
- It seems that SAGA-SZHT is missing in both the main text and appendix. Given its good performance in Table 1, I am a bit confused on why it is not properly introduced. Are there some errors?
- Regarding experimental results, I am a bit confused by Figure 3. It seems that SZOHT converges faster than all other methods, but gradually becomes worse with more iterations. Some explanations may be needed here on why we need so many iterations after convergence.
- Also, the evaluation of few-pixels universal adversarial attacks may seem a bit restricted with only 10 images from the same class. It would be better if the authors can use more images from different classes to further justify the effectiveness of proposed method.

---

> ### Author Response · Authors · 2023-11-21
> **Response to Reviewer wexC [1/3]**
>
> Dear reviewer wexC, thank you for your insightful review. We sincerely hope that the main concerns raised in the review can be clarified by the following responses.
>
> > **Your comment:** My first concern is on the usefulness of zeroth-order oracles. From experiments in this paper, it seems that we have easy access to gradient information in all these applications. Therefore, why do we need to emphasize zeroth-order algorithms? The authors may consider some other applications where gradient information is hard to obtain.
>
> **Our response:** Thank you for your question. First, we would like to highlight that our experiments on universal adversarial attacks are in the black-box setting: such attacks are a real-life example of setting where one does not have access to the gradient of the objective function. Indeed, in a typical setting, one would seek to attack a machine learning service which could only be accessed through an API, processing the input image and returning, say, the probabilities of each class for the image, which is exactly the inputs and outputs which we consider in our task.
> Second, regarding our experiments with ridge regression, we have actually considered them as they are a very common task in the optimization literature, since those cost functions verify the (restricted) strong convexity and (restricted) smoothness assumptions, and as such, they can illustrate precisely the performance of our algorithms and verify the validity of our theoretical results. However, it is worth noting that, even for such experiments, one could still think of a setting where gradients would be inaccessible: the setting of private distributed learning. Indeed, in one variant of such setting, a central server is only allowed to receive a scalar value from each of the workers: one way to address such setting is for each worker to send the value of its directional derivative for a given random direction: there, the full gradient is therefore inaccessible to the central server, who can only construct a zeroth-order approximation of it (see e.g. [4], [5]).
>
>
> > **Your comment:** Another concern is on the setting of L0 regularization. While there exist many other approaches on obtaining sparse solutions (e.g., LASSO), the authors may need to justify the necessity of using L0 regularization here.
>
>
> **Our response:** Thank you for your question, we believe that the main advantage of pursuing $\ell\_0$ optimization directly through the hard-thresholding operator rather than using convex relaxations (such as $\ell\_1$ optimization) is regarding the control that one has on the sparsity of the iterates. Indeed, ensuring a specific level of sparsity $k$ is very tedious with $\ell\_1$ regularization: one has to tune the strength of the regularization $\lambda$ through an extensive grid-search, to find the $\lambda$ which gives a solution of sparsity exactly $k$. On the other hand, when using hard-thresholding, one only needs to specify $k$ at the beginning of the algorithm, and the algorithm can then be run once, without any grid-search. Specifying a given $k$ is important in various real-life tasks, for instance the few-pixels adversarial attacks task that we consider in our experiments (Sec. 5): there, one typically seeks to attack an image using a given number $k$ of perturbed pixels. A second example is in portfolio optimization, where one, due to budget limitations, one is only allowed to invest in a maximum of $k$ examples to optimize its return [6] . Finally, a last example is in compressed sensing, where one seeks to recover a vector of sparsity known to be exactly $k$ [7]. In all of these examples, hard-thresholding algorithms are especially adapted to the task, as they can control such $k$ explicitly.
>
> > **Your comment:** I am a bit confused on the key contribution of this work. The authors introduced two algorithms, pM-SZHT and VR-SZHT. What is the key difference of these two algorithms? Moreover, pM-SZHT is not compared in experiments, but SARAH/SAGA-SZHT is compared. This is also very confusing and may need some explanations.
>
>
> **Our response:** We apologize for the lack of necessary explanations and have made revisions to the rebuttal version. In our convergence analysis using the p-Memorization (pM) framework, which includes many variance reduce algorithms, we define the update method for parameter $a$ in pM-SZHT as follows:
>
> $$
> \hat a^+\_j:=\begin{cases}
> \hat\nabla f\_j(\theta),  \textrm{if} ~  j \in J \\\
> \hat{a\_j}, \textrm{otherwise}
> \end{cases}
> $$
>
> Regarding the difference between VR-SZGT and pM-SZHT, actually, our paper can cover two possible variants of VR-SZHT, the first one being an instantiation of pM-SZHT, and which can be described as follows: we fix $p > 0$ and draw in each iteration $r \sim
> U[0; 1)$. If $r < p/n$, a complete update, $\hat a^+\_j=\hat\nabla f\_i(\theta),\forall j$ is performed, otherwise, they are left unchanged. **[...Continued below...]**

---

> > ### Author Response · Authors · 2023-11-21
> > **Response to Reviewer wexC [2/3]**
> >
> > **[...Continuing from above...]** Such an algorithm indeed fits in the pM-SZHT framework and resembles the original VR-SZHT algorithm (see Hofmann et. al. 2015).
> > As such, the analysis of such variant of VR-SZHT can be deduced directly from the analysis of pM-SZHT. However, this variant typically does not have good performance. Therefore, to provide a comprehensive analysis of all variance reduction algorithms used in practice, we actually provide in our paper the analysis of the standard VR-SZHT, i.e. the ZO hard-thresholding version of the original SVRG algorithm defined in Johnson et. al. (2013).
> > Regarding SARAH, it is a variance reduction method independent of the pM-SZHT framework. Due to space limitations, we have analyzed it and included this section in the Appendix (we have added the reference to such Appendix Section in the main body of the new revision, in the experimental section), thank you for your suggestion.
> >
> >
> > > **Your comment:** It seems that SAGA-SZHT is missing in both the main text and appendix. Given its good performance in Table 1, I am a bit confused on why it is not properly introduced. Are there some errors?
> >
> > **Our response:** As we briefly mention at the end of page 5, SAGA-SZHT is actually a specific instantiation of pM-SZHT, where we only need to let $J \in \\{\\{1\\}, ..., \\{n\\}\\}$ and $p=1$. Thank you for your comment, we agree that this point was not highlighted enough: in the new revision, we also included a bibliographic reference to SAGA at the beginning of page 6 for clarity. Additionally, we did not separately provide a proof of convergence for SAGA-SZHT since it falls within the pM-SZHT framework, but we used SAGA as our main example to analyze algorithms within the pM framework.
> >
> >
> > > **Your comment:** Regarding experimental results, I am a bit confused by Figure 3. It seems that SZOHT converges faster than all other methods, but gradually becomes worse with more iterations. Some explanations may be needed here on why we need so many iterations after convergence.
> >
> > **Our response:** Thank you for your question. First, we give a slight precision regarding our experimental protocol: we have now updated our experiments in the new revision: before, we kept the last iterate at the end of training as our final model for attacking the images (and ran our grid-search according to it), but it makes indeed more sense to rather keep the best iterate (and update the grid search accordingly): this is what we do now in the new revision: as such, even for algorithms that worsen over time, we will keep the best iterate along the curve, which is fairer. Note that the reason why the adversarial experiments are more erratic, with some potential worsening of the objective function along training, is that adversarial attacks actually do not verify the theoretical assumptions in our paper, since, they are not smooth problems and, a priori, are not RSC either. Therefore, the behavior of our method on such problems is harder to predict: however, we provide such experiments for indicative purposes, as it is an interesting use case for our algorithms. Note however that in the new revision we have ran more intensive experiments, which show that still, even in such setting, variance reduction methods (in particular, SAGA-SZHT and SARAH-SZHT) perform well compared to SZOHT and FGZOHT (as can be seen in the updated main body revision of the adv. attacks section, and the extra results in the Supplemental, see also the response below).
> >
> >
> > > **Your comment:** Also, the evaluation of few-pixels universal adversarial attacks may seem a bit restricted with only 10 images from the same class. It would be better if the authors can use more images from different classes to further justify the effectiveness of proposed method.
> >
> > **[...Continued below...]**

---

> ### Author Response · Authors · 2023-11-21
> **Response to Reviewer wexC [3/3]**
>
> **[...Continuing from above...]**
>
> **Our response:** Thank you, following your suggestion, we have run more experiments on the adversarial attacks setting: we considered 3 more classes: ‘airplane’, ‘ship’, and ‘bird’ (in addition to the original ‘dog’ class) which results we provide in the revised supplemental (note that we have now put the class ‘dog’ instead in Appendix, and replaced it by ‘airplane’ in the main body): those confirm even further the advantage of variance reduction in ZO hard-thresholding: in Figure 3 one can see that in terms of objective value, SARAH-SZHT and SAGA-SZHT outperform all other algorithms. And in terms of attack success rate (Table 1), SARAH-SZHT outperforms other algorithms (by successfully attacking 7/10 images). Similar observations can be done on the other classes (‘ship’, ‘bird’, and ‘dog’), which we show in the revised Appendix. Regarding the small number of images per class, and the fact that for each experiment, images are of the same class, this constraint comes from the fact that we consider universal adversarial attacks rather than vanilla adversarial attacks. More precisely, since one same perturbation must be used to attack all images at once, taking more images or images of mixed classes would make the task much harder and erratic in practice. Therefore, we follow the protocol of taking a small batch of images of the same class, which is what is usually done in the litterature (see e.g. Appendix A.11 in [8]).
>
>
> [4] Zhang, Qingsong, et al. Desirable companion for vertical federated learning: New Zeroth-order gradient based algorithm. Proceedings of the 30th ACM International Conference on Information \\& Knowledge Management, 2021
>
> [5] Gratton, Cristiano, et al. Privacy-preserved distributed learning with zeroth-order optimization. IEEE Transactions on Information Forensics and Security, 2021
>
> [6] Bertsimas, Dimitris, and Ryan Cory-Wright. A scalable algorithm for sparse portfolio selection. Informs journal on computing, 2022
>
> [7] Foucart, Simon, et al. An invitation to compressive sensing. Springer New York, 2013.
>
>
>  [8] Sijia Liu, Zeroth-order stochastic variance reduction for nonconvex optimization. NIPS, 2018

---

### Official Review · Reviewer_iAcw · 2023-11-01

**Soundness:** 3 good
**Presentation:** 3 good
**Contribution:** 2 fair
**Rating:** 6
**Confidence:** 3

**Summary:**

The paper provides a new analysis of SZOHT and provides a new perspective on conflict of zeroth-order methods and hard thresholding, then introduces variance reduction to improve convergence.

**Strengths:**

1. The generalized algorithm framework and analysis are comprehensive, demonstrating the role of variance in convergence guarantee.
2. The introduction of variance reduction into SZOHT is new and query complexity is reduced.

**Weaknesses:**

1. I don't see major weaknesses of the work, maybe more experiments on larger scale problems would further validate the practical benefits of variance reduction.

**Questions:**

Out of curiosity, when applying variance reduction to SZOHT, are there any major technical difficulties in the analysis or major new techniques used in the work?

---

> ### Author Response · Authors · 2023-11-21
> **Response to Reviewer iAcw**
>
> Dear reviewer iAcw, we extend our sincere gratitude for dedicating your valuable time to reviewing our paper and for expressing your appreciation and support for our work. In the following, we will provide a comprehensive response to your review comments.
>
> > **Your comment:** Out of curiosity, when applying variance reduction to SZOHT, are there any major technical difficulties in the analysis or major new techniques used in the work?
>
>
> **Our response**: Indeed, in our work which applies variance reduction techniques to zeroth-order hard-thresholding, there are several technical challenges that arise, which necessitate the use of new techniques, departing from traditional techniques in variance reduction from both the first-order IHT case, and the zeroth-order convex case, as we describe below:
>
> -   **New Perspective on Resolving Conflicts Between Zeroth-Order Methods and Hard-Thresholding.** First, the conflict between hard-thresholding and the ZO gradient estimate is unique to the specific combination of (i) ZO and (ii) hard-thresholding (HT), since the ZO error is related to the sparsity of the iterates ($k$) (contrary to, say, the error of first-order algorithms, which is related to the batch-size, independent from $k$), and such $k$ already has specific constraints dictated by the convergence analysis from Iterative Hard Thresholding. As such, analyzing how variance reduction reduces the error in the ZO gradient is very specific to the combination of (i) ZO and (ii) HT, and results in particular in a novel multi-parameter analysis, which we describe below.
>
>
> - **Multi Parameter Analysis.** Understanding parameter relationships is vital in convergence analysis, especially in explaining how variance reduction techniques alleviate conflicts. Here, we illustrate this point by Comparing SZOHT and VR-SZHT.  In SZOHT, attention is primarily on the constraint link between the zeroth-order estimated direction number $q$ and the Hard Thresholding parameter $k$. However, VR-SZHT introduces variance reduction, adding an inner loop parameter $m$, leading to interdependence among these three parameters. Analyzing this relationship poses a notable challenge. To assess if variance reduction algorithms can mitigate conflicts, we explore whether there exists $m$ to make the algorithm converge for any $q$. Analyzing the coefficients in the convergence term of equation (8) yields inequalities involving $k$,$q$,$m$. Ultimately, confirming a positive outcome is achieved through the existence of $m$. For additional insights, refer to Appendix 4.3.
>
> - **General Analysis:** Finally, the introduction of a general analysis framework is another new technique used in the paper. This framework systematically assesses the performance and behavior of diverse variance-reduced algorithms under $\ell\_0$ constraints and zeroth-order gradient scenarios. Additionally, we have substantiated the theoretical basis for various common VR methods.

---

### Official Review · Reviewer_TcW5 · 2023-11-02

**Soundness:** 2 fair
**Presentation:** 2 fair
**Contribution:** 3 good
**Rating:** 5
**Confidence:** 3

**Summary:**

The paper presents a new algorithm for solving sparse learning problems in machine learning. By incorporating variance reduction techniques, the algorithm improves convergence rates and expands its applicability. It offers a solution to conflicts between zeroth-order gradients and hard-thresholding and provides a general analysis framework. Overall, the paper contributes an efficient and effective approach for sparse learning.

**Strengths:**

1. This paper introduces an interesting perspective, i.e., variance reduction, to improve existing zeroth-order hard-thresholding algorithms.
2. This paper has provided sound theoretical analysis for the newly proposed algorithm.

**Weaknesses:**

1. There are many symbol error as well as misleading notations in the paper, which requires further correction. E.g., the $k^*$ should be s in Assumption 1; no introduction of $\mu$ in eq. 3; no explanation for $H_{2k}$ in page 3 when it appears at the first time; what's the meaning of J in page 5? how to get the i in page 5? and so on. This will make it hard for readers to understand the paper.
2. More details and discussion for experiment section, e.g., the explanation on IZO and NHT in Fig.2,3, how to tell VR-SZHT is better in Table 1?
3. More experiments may do a better job in supporting the advantages of the VR-SZHT algorithm. Currently, VR-SZHT seems to be even worse than baselines in some cases, e.g., Fig. 3.

PS: it's really hard for reviewers to point out which equation when there is no equation number!

**Questions:**

1. Could the authors elaborate more on what's the conflict between the expansionary of hard-theresholding and ZO error about? And how it affects the performance of zeroth-order hard-thresholding algorithms?

---

> ### Author Response · Authors · 2023-11-21
> **Response to Reviewer TcW5**
>
> Dear reviewer TcW5, thank you for your insightful review. We sincerely hope that the main concerns raised in the review can be clarified by the following responses.
>
> > **Your comment:** There are many symbol error as well as misleading notations in the paper, which requires further correction. E.g., the $k^\*$ should be $s$ in Assumption 1; no introduction of $\mu$ in eq. 3; no explanation for $H\_{2k}$ in page 3 when it appears at the first time; what's the meaning of J in page 5? how to get the i in page 5? and so on. This will make it hard for readers to understand the paper.
>
> **Our response:**
> Thank you for pointing out these missing notations and precisions, following your suggestions, we have fixed $k^\*$, introduced $\mu$, $\mathcal{H}\_{2k}$, $J$, and $i$.
>
>
> > **Your comment:** More details and discussion for experiment section, e.g., the explanation on IZO and NHT in Fig.2,3
>
>
> **Our response:**  Thank you for your suggestion, we have added the definition of IZO and NHT in the new revision.
>
> > **Your comment:** How to tell VR-SZHT is better in Table 1? More experiments may do a better job in supporting the advantages of the VR-SZHT algorithm. Currently, VR-SZHT seems to be even worse than baselines in some cases, e.g., Fig. 3
>
> **Our response:** In the new revision, we ran more experiments (3 more classes for the adv. attacks), and those show more clearly the advantage of variance reduction algorithms: we now emphasize the class ‘airplane’ in our revised main body (Figure 3 and Table 1, updated) (and we kept the previous ‘dog’ example in the revised Appendix with the two added examples): in such ‘airplane’ case, we can clearly see the advantage of SARAH-ZHT in the table: it can successfully attack 7 out of the 10 images: this is even further confirmed on the other adversarial attacks settings that we added in the revised supplemental. Regarding VR-SZHT, in some experiments it can be indeed worse than some baselines (perhaps due to instabilities in adversarial attacks, which do not respect the RSC-RSS assumptions and are given for indicative purposes), but it can be seen to perform better than its vanilla stochastic counterpart (SZOHT) e.g. in Figures 5 and 6 in Appendix.
>
> > **Your comment:** Could the authors elaborate more on what's the conflict between the expansionary of hard-theresholding and ZO error about? And how it affects the performance of zeroth-order hard-thresholding algorithms?
>
> **Our response:** Thank you for your question, we hope to provide here some more explanations on the conflict between expansivity of the hard-thresholding operator, and the errors of the ZO gradient estimator. For clarity, we first start with common knowledge from convex optimization, and then describe why a particular challenge arises when combining zeroth-order and hard-thresholding. From an intuitive perspective, zeroth-order convex constrained optimization convergence relies on a fundamental property: the non-expansivity of the projection operator. Such property ensures that projecting $\theta\_t' := \theta\_t - \eta g(\theta\_t)$ onto the constraints can only make $\theta\_t'$ closer to the optimum. In such case, even if the error of the gradient estimator is high, one can always set the learning rate to some small enough value, to avoid instabilities and ensure (in expectation) convergence of the algorithm (even if such convergence is slow). But in $\ell\_0$ optimization, the hard-thresholding operator (i.e. projection onto the $\ell\_0$ pseudo-ball), is *not* non-expansive. As such, projecting a vector $\theta\_t'$ onto the $\ell\_0$ pseudo-ball can bring it *further away* from the optimum. In such case, setting a small learning rate (when the error in gradient estimation is high) may be detrimental to the convergence, since iterates can get further and further away from the optimum. There, one must actually ensure that enough progress is made by $\theta\_t - \eta g(\theta\_t)$ towards the optimum, so that even after hard-thresholding, iterates still get closer (in expectation) to the optimum. As such, the learning rate cannot be taken arbitrarily small, and the gradient error therefore needs to be tackled directly, e.g. by sampling enough random directions, and/or using variance reduction techniques as we propose in our paper.

---

### Official Review · Reviewer_5xEJ · 2023-11-05

**Soundness:** 3 good
**Presentation:** 3 good
**Contribution:** 3 good
**Rating:** 6
**Confidence:** 2

**Summary:**

This paper  provides a new insight into variance reduction: mitigating the unique conflicts between ZO gradients and hard-thresholding. Under this perspective, this paper proposes a generalized variance reduced ZO hard-thresholding algorithm as well as the generalized convergence analysis under standard assumptions. The theoretical results demonstrate the new algorithm eliminates the restrictions on the number of random directions, leading to improved convergence rates and broader applicability compared with SZOHT.

**Strengths:**

The theoretical results demonstrate the new algorithm eliminates the restrictions on the number of random directions, leading to improved convergence rates and broader applicability compared with SZOHT.

**Weaknesses:**

Typo:
It should be $\| \nabla f_i(\theta) - \nabla f_i(\theta') \| \leq \rho_s^+ \|\theta - \theta'\|$ in the Assumption 2.

**Questions:**

Since the Problem (1) is NP-complete, what  is $\theta^*$ in the convergence analysis? If $\theta^*$ is the optimal point of Problem (1), why can the proposed algorithm can achieve a  linear convergence rate which means it is a polynomial time algorithm that can solve a NP-complete problem?

---

> ### Author Response · Authors · 2023-11-21
> **Response to Reviewer 5xEJ**
>
> Dear reviewer 5xEJ, we extend our sincere gratitude for dedicating your valuable time to reviewing our paper and for expressing your appreciation and support for our work. In the following, we will provide a comprehensive response to your review comments.
>
> > **Your comment:** Typo: It should be $\\|\nabla f\_i(\theta) - \nabla f\_i(\theta')\\| \leq \rho\_s^+ \\| \theta - \theta' \\|$  in the Assumption 2.
>
> **Our response:** Thank you for your correction, we have fixed such typo.
>
> > **Your comment:** Since the Problem (1) is NP-complete, what is $\theta^\*$
>  in the convergence analysis? If $\theta^\*$
>  is the optimal point of Problem (1), why can the proposed algorithm can achieve a linear convergence rate which means it is a polynomial time algorithm that can solve a NP-complete problem?
>
>
>
> **Our response:** Thank you for your question. Since the problem is NP-hard, we indeed do not prove convergence to optimality in polynomial time, and there are two aspects under which our results depart from convergence to optimality.
>
> 1. First, our convergence result bound relates $\mathcal{F}(\theta\_t)$ for our iterates $\theta\_t$ of sparsity $k = \Omega(\kappa^2 k^\*)$ (with $\kappa = \frac{\rho\_s^+}{\rho\_s^-}$), to $\mathcal{F}(\theta^\*)$ where $\theta^\*$ is of sparsity $k^\* < k$ and can be taken as $\theta^\* := \arg\min\_{\|\theta\|\_{0} \leq k^\*} \mathcal{F}(\theta)$ (by comparison, a typical convex optimization bound would be written for a $\theta^\*$ of sparsity $k$ (i.e. both the minimum and the iterates would belong to the same constraint set, which is not the case here for us)). Such kind of results may be non-standard from a convex optimization perspective, but it is how all the results in the Iterative Hard Threholding (IHT) literature with RSC and RSS assumptions, are formulated (see e.g.  Jain et al. (2014), Nguyen et al. (2017), Li et al. (2016), Zhou et al. (2018), and de Vazelhes et al. (2022)). In fact, it was actually shown in [3] that the factor $\kappa^2$ of the sparsity relaxation cannot be improved in the analysis of IHT.  Finally, we highlight that usually hard-thresholding methods are used for very high-dimensional problems, and as such, even relaxing the sparsity by a constant factor provides sparse enough solutions.
>
> 2. Second, our result exhibit a system error, i.e. we prove results of the form $R(\theta\_t) \leq R(\theta^\*) + \varepsilon + \Delta\_1 \mu^2 + \Delta\_2$ , where $\Delta\_2$ is a constant system error (e.g. in Theorem 2, it is a consequence of the constant $L\_r$). Such system error is also standard in IHT results, see e.g. Li et. al. 2016 (Theorem 3.3), Nguyen et. al. (2014), de Vazelhes et. al. (2022). It is also worth mentioning that when $\mathcal{F}$ has a $k^\*$-sparse unconstrained minimizer, which is often the case in practical applications such as sparse reconstruction or when considering overparameterized deep networks [2], such system error vanishes. And even if $\mathcal{F}$ does not have such minimizer, this error is usually acceptable.)
>
>
>
>
>
> [1]Prateek Jain, Ambuj Tewari, and Purushottam Kar. On iterative hard thresholding methods for high-dimensional m-estimation. In Advances in Neural Information Processing Systems, volume 27, 2014.
>
> [2]Alexandra Peste, Eugenia Iofinova, Adrian Vladu, and Dan Alistarh. Ac/dc: Alternating compressed/decompressed training of deep neural networks. Advances in Neural Information Processing Systems, 34, 2021.
>
> [3]Axiotis, Kyriakos, and Maxim Sviridenko.Iterative hard thresholding with adaptive regularization: Sparser solutions without sacrificing runtime. International Conference on Machine Learning. PMLR, 2022

---

### Author Response · Authors · 2023-11-21
**Global Rebuttal**

We would like to thank all the reviewers for their insightful comments and useful suggestions. The main concern revolves around clarifying the presentation and the related work, as well as the contribution and the challenges in applying variance reduction techniques to the hard-thresholding zeroth-order setting, which we sincerely hope can be addressed by the following brief summary of the main contributions of our paper. Our work tackles the unique conflict arising from the specific combination of hard-thresholding (HT) and zeroth-order (ZO) (due to the fact that both the ZO error and the HT expansivity depend on the sparsity $k$ of the iterates), and provides a comprehensive analysis of several variance reduction algorithms in such setting. Through an in depth-analysis of the parameters of such variance reduction algorithms and  how they interact with the aforementioned conflict between gradient error and hard-threshoding, we can prove that variance reduction techniques can indeed successfully reduce the variance of the gradient estimator, to the point that they can even eliminate the stringent requirement of having a minimum number of random directions $q$, which was present in the previous work of de Vazelhes et al. (2022).

Additionally, we have carefully revised the manuscript based on the reviews. The following is a summary of major changes (we have colored our modifications in red in the revised documents for convenience):

- We added a more intensive experimental analysis, by running our universal adversarial attacks on three additional classes of the CIFAR-10 dataset, and providing the detailed curves and attacked images for such settings, further confirming the applicability of variance redution technique for zeroth-order hard-thresholding in practice.
- In the main body of the revision, we have added the necessary reference to the works from the literature for the variance reduction algorithms, as well as more references to the appropriate sections of the Appendix, for sake of clarity.
- Additionally, we added an extra section in Appendix which sums up, in a table, all variance reduced algorithms, their original first-order counterpart and computational complexity, and links to the corresponding section in the paper where we analyze them.

---

### Meta-Review · Area_Chair_3fNY · 2023-12-05

**Metareview:**

The reviewers have some spread of scores; however, they appreciate the technical insights, contributions both theoretical and experimental, and the great lengths the authors have gone to enhance the manuscript and address all concerns during the discussion phase. Moreover, the contributions of the paper are relatively more unique, which means they may be a valuable addition to ICLR technical proceedings. For this reason I suggest this paper be accepted.

**Justification For Why Not Higher Score:**

NA

**Justification For Why Not Lower Score:**

NA

---

### Decision · Program_Chairs · 2024-01-16

Accept (poster)